# Land Use and Landscape Characteristics Are Associated with Core Forest Patches in Ghana

**Joseph Oduro Appiah** [1,*] , **Dina Adei** [2] and **Williams Agyemang-Duah** [3]

1   Department of Geography, Environment & Spatial Analysis, Cal Poly Humboldt, Arcata, CA 95521, USA
2   Department of Planning, Kwame Nkrumah University of Science and Technology, Kumasi P.O. Box 80404, Ghana
3   Department of Geography and Planning, Queen's University, Kingston, ON K7L 3N6, Canada
*   Correspondence: jo216@humboldt.edu; Tel.: +18-62-2359-994

**Abstract:** Land uses and terrain characteristics would likely influence the types and spatial arrangements of forest patches, and generally, forest fragmentation. Whereas prior research has focused mainly on direct land use-induced forest fragmentation, this study models the relationship between the spatial distribution of core forest patches, land uses, and terrain variables. Relying on Landsat images from the Atewa Range Forest Reserve (ARFR) in Ghana, we use machine learning geospatial techniques and statistical methods to process satellite images and model the relationship between core forest patches and associated variables. The study finds that a unit reduction in elevation would significantly likely reduce by 0.995 times the possibility of forest patches being core forests, implying that on lower slopes, core forests are less likely to occur. Additionally, we find that a unit increase in slope gradient significantly increases the odds of a forest patch being among the core forest category by 1.35 times. Moreover, our results show that the odds of forest patches being core forests significantly increase by 1.60 and 2.14 times if patches are found beyond 1 km from logging sites and access roads, respectively. This implies that intact forest patches would likely be found on higher slopes, higher elevations, and areas far away from land uses. Based on the results, we suggest that the protection of forest patches should target higher elevations and slopes and most importantly areas far from land uses whereas forest restoration programs should target areas close to land uses and on lower elevations and lower slopes. With this study demonstrating a significant relationship between core forests, land uses and terrain variables, we present important information to land managers for land monitoring and conservation in the ARFR and other tropical forest regions of the world.

**Keywords:** forest patches; random forest; environmental modeling; logistic regression

## 1. Introduction

Anthropogenic land uses contribute significantly to global environmental change [1–4]. Over the past few decades, land uses have contributed to the increasing forest cover degradation, a phenomenon that is likely to contribute more to climate change in the next years ahead [5,6]. Governmental and non-governmental organizations are contributing to combating forest degradation [7–10]. However, there is still some more work to be done to reduce the rate of forest degradation, especially in tropical regions where the exploitation of forest resources is a greater part of people's livelihoods [11–13]. Nonetheless, the degradation of the forests in the quest to satisfy livelihood needs would likely adversely impact livelihoods in the future.

Core forest, an area of forest occurring outside the edge effect area and is not degraded by fragmentation [14,15], would likely perform socio-ecological functions (Note: the edge effect area is a distance of 100 m from non-forest to the forest interior where human activities, wildfires, micro-climate effects, and the effects of fragmentation would likely

degrade forest cover [16,17]). Maintaining intact and large contiguous core forest patches would be necessary for preserving the integrity of forest ecosystems, providing non-timber forest products for communities, and reducing the impacts of global climate change through carbon sequestration [18,19]. Thus, the protection of a larger part of the core forest from land uses would be necessary, and to do so, there is a need to study the association between a variety of land uses and core forest patches. This would improve land managers' understanding of the various land uses that would likely contribute to the degradation of core forest patches and consequently develop policies and management measures for the protection of the core forests. Moreover, this would ensure that policies and management initiatives that protect the forests are created in such a way that both ecological integrity and sustainable land use are ensured.

Our study area is the Atewa Range Forest Reserve (ARFR). Focusing our study on the ARFR, an area that falls within the tropical forest region, offers room to compare our study findings with findings from other tropical forest regions of the world. Similar to other tropical forest environments where land uses are increasing, the ARFR is experiencing a recent increase in the rate at which forest cover is being cleared for various land uses [20–22]. The situation of forest degradation in the forest would likely increase with the proposition of the site for large-scale bauxite mining [23]. For instance, previous studies have noted that encroachment by farmers and recent mining exploration are on the rise and have contributed to disturbances in the forest ecosystem of the reserve [22,24,25]. The various land uses and forest cover patches would likely be influenced by terrain characteristics (e.g., slope and elevation). For instance, intact forest cover patches would likely be found on higher elevations due to a lack of access by humans. The ARFR as a protected landscape and a globally significant biodiversity area (GSBA) is expected to have a minimal amount of human influence, and the landscape is given the best possible protection while ensuring that non-threatening anthropogenic activities (e.g., harvesting non-timber forest products) are undertaken sustainably.

Previous land use and land cover studies in tropical forest regions (e.g., Asia, Africa, and Latin America) have focused mostly on measuring and predicting direct land cover change and fragmentation [21,26–29]. However, similar to predicting direct land cover change, Oduro Appiah and Agyemang-Duah [30] used geospatial approaches and statistical modeling to measure the relationship between land uses and forest patch sizes in the Tano-Offin Forest Reserve in Ghana. In the ARFR, Kusimi [22] used GIS and remote sensing approaches to measure land use and cover change, noting how land use impacts forest cover. These studies in the tropical regions contribute to understanding the dynamics of land uses and most importantly, emphasize the relevance of applying geospatial approaches in measuring landscape characteristics. However, measuring the relationship between core forest patches, land uses, and other associated factors was beyond the scope of the previous studies. Thus, this study builds on previous studies by filling this knowledge gap.

Based on the knowledge gap identified, the objective of this study is to propose a model that expresses the relationship between core forest patches (dependent variable), land uses, and terrain characteristics (independent variables). The study uses GIS and remote sensing techniques (random forest machine learning classification) and methods from landscape ecology to process satellite images and construct the patches of forest cover and other land uses. With the aid of a logistic regression model, we propose a model that shows the association between core forest patches, land uses, and terrain characteristics (e.g., slope and aspect). Our study is relevant because it provides a spatial model for landscape managers and policymakers to develop land management strategies needed for protecting core forest patches from being impacted by the footprints of anthropogenic activities. Most importantly, with the outcome of this study, land managers can develop measures to ensure that anthropogenic activities are undertaken sustainably.

## 2. Materials and Methods

### 2.1. Description of the Study Area

The ARFR is located within 05°58' to 06°20' N and 0°31' to 0°41' W, and it is surrounded mostly by agrarian communities (see Figure 1). According to Lindsell et al. [31], the ARFR is approximately 263 km$^2$ (26,300 ha). Most importantly, the ARFR is part of the tropical rain forest, a stretch of land found across three continents, including South America, Africa, and Asia. Being within the tropical rainforest region is an indication that the ARFR receives frequent and high amounts of rainfall, and such environmental condition plays a relevant role in the growth of plants. The ARFR contains about 656 vascular plant species, including 323 species of tree, 83 types of shrub species, 155 climber and liane species, 68 species of herbaceous plants, 22 types of epiphytes, and 5 grassland species. It houses about 200 birds and 700 species of butterfly fauna [32]. Due to its ecological importance, the ARFR has been classified as an Important Bird Area (IBA) and a GSBA [20].

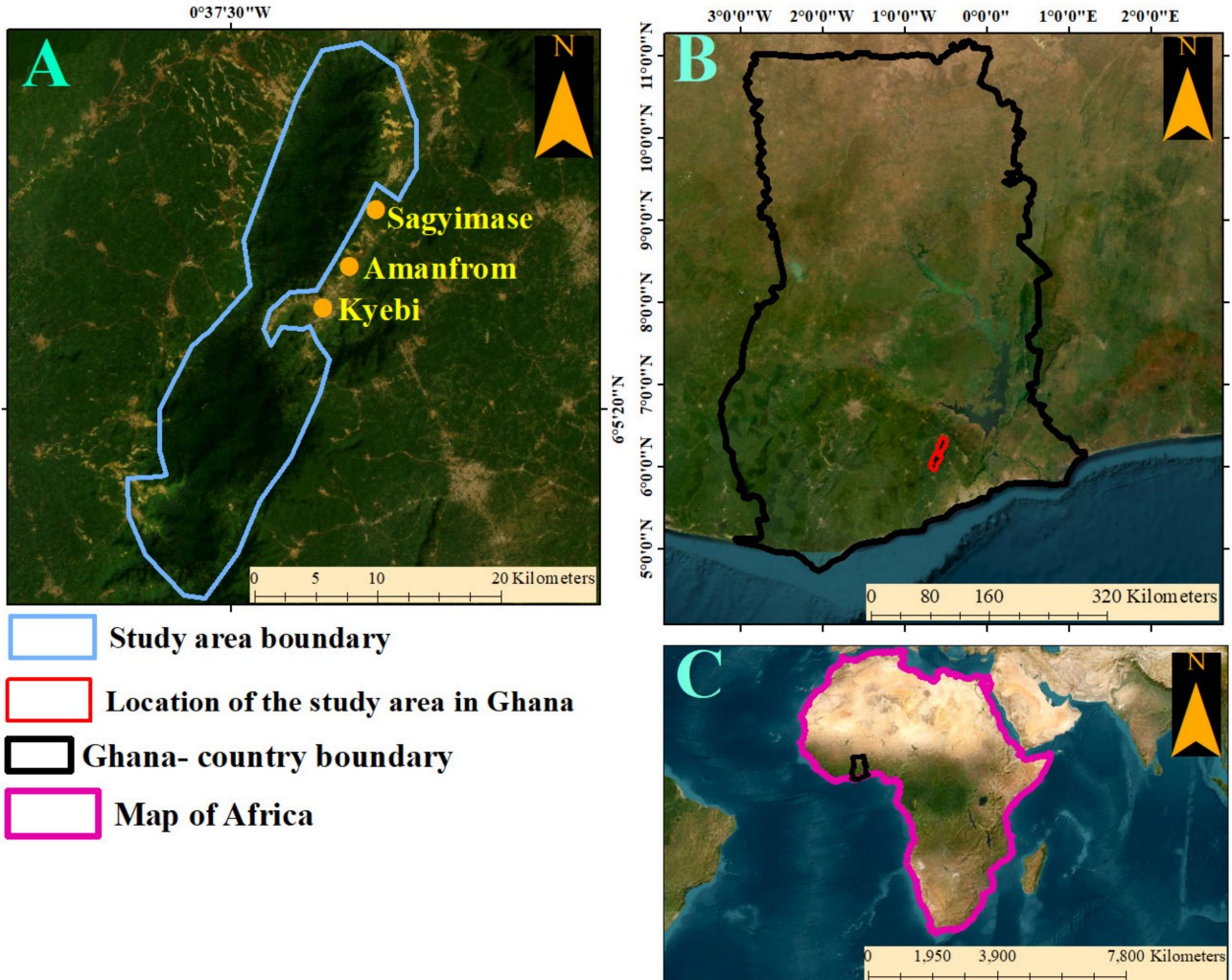

**Figure 1.** Study area location. (**A**) shows the area covered by the Atewa Range Forest Reserve, (**B**) shows the study area in the context of Ghana, and (**C**) shows Ghana in the context of Africa.

Whereas the protected status has been able to reduce forest degradation to some extent, the rate of degradation has been faster over the past few decades. For instance, Meijer et al. [20] have noted that the closed-canopy forest cover reduced from 88% in 1990 to 60% in 2010. Similarly, the closed-canopy forest cover formed about 91% of the total reserve area but decreased to about 81% in 2010. Most of the closed-canopy forest cover

degradation occurred after 2000, and this has mostly given rise to open-canopy forest cover. Whereas the ecosystem has been under continuous threat from human activities (such as agriculture) over the past few decades, potential threats from large-scale mining and quarrying activities within the forest reserve boundaries are issues of concern in the years after 2000 [20]. Nonetheless, the Forest Protection Act (Amendment from Forest Protection Decree of 1974), 2002 (Act 624) highlights the prohibited human activities to protect the integrity of forests, including the ARFR. This law was intended to prevent people from farming (pastures, cultivating [food and cash] crops), lumbering (cutting trees for timber), obstructing the channel of any water bodies, building, hunting, and setting fire to forests (Parliament of the Republic of Ghana [33] cited in Oduro Appiah et al. [34]. Regardless of the law taking effect in 2002, the degradation of the forest, as noted earlier, has intensified within the same decade the law was passed.

## 2.2. Data Collection

We made use of one of the most recent Landsat images downloaded from the United States Geological Survey (USGS) archive. As indicated by the USGS, the image is from Operational Land Imager (OLI)/Thermal Infrared Sensor (TIRS) 8, and USGS acquired the image on 2 January 2020, from path 193 and row 056. The Landsat image bands used in this study were band 2—blue, band 3—green, band 4—red, band 5—near-infrared, band 6—shortwave infrared 1, band 7—shortwave infrared 2, and band 8—panchromatic. Whereas bands 2–7 were 30 m by 30 m in spatial resolution, band 8 was 15 m by 15 m in spatial resolution. The image is part of the tier 1 surface reflectance image and thus, it has already been geo-rectified and radiometrically corrected. We processed the Landsat image to acquire information about the forest cover patches in the ARFR.

Apart from the Landsat image, we also made use of high-resolution satellite images from Google Earth Pro. These high-resolution images had spatial resolution ranging between 1 m and 1.5 m, and on Google Earth Pro, the image has been dated 13 January 2020. Most of the spatial data for land uses (e.g., access roads, mine sites, human settlement) were acquired from high-resolution images. Apart from the satellite images, we used digital elevation model (DEM) data. The DEM data were acquired from the USGS using the Global Multi-resolution Terrain Elevation Data (GMTED) 2010 search engine. We selected the DEM with 7.5 arc seconds spatial resolution to reduce the USGS processing errors (root mean square errors [RMSE] 26–30 m) and their impacts on our analysis. Elevation, slope, and aspect were derived from the USGS DEM.

## 2.3. Satellite Image and Digital Elevation Model Processing

Bands 2–7 of the Landsat image were composited for land cover classification. However, the composited image was pan-sharpened with the panchromatic band (band 8) to create a 15 m resolution image for further processing. The pan-sharpened image was classified using the Random Forest (RF) machine learning classification algorithm [35]. We used the RF algorithm because of the following reasons. First, Rodriguez-Galiano et al. [36] have noted that the main advantages of RF include but are not limited to their non-parametric nature, high classification accuracy, and ability to determine variable importance. Second, the RF as a non-parametric algorithm has been used many times in remote sensing studies because it handles high data dimensionality and multicollinearity successfully [37]. In this study, with the aid of ensembles of decision trees [36,37], classes of land use and land cover were determined based on majority votes from all trees. The image was classified into forest cover (evergreen, deciduous, and semi-deciduous trees), agricultural land (cropland, herbs, and bushes), developed land (built-up area, mining, and logging sites), and water (ephemeral and permanent rivers, lakes, ponds, etc.) (see Figure 2 Panel A).

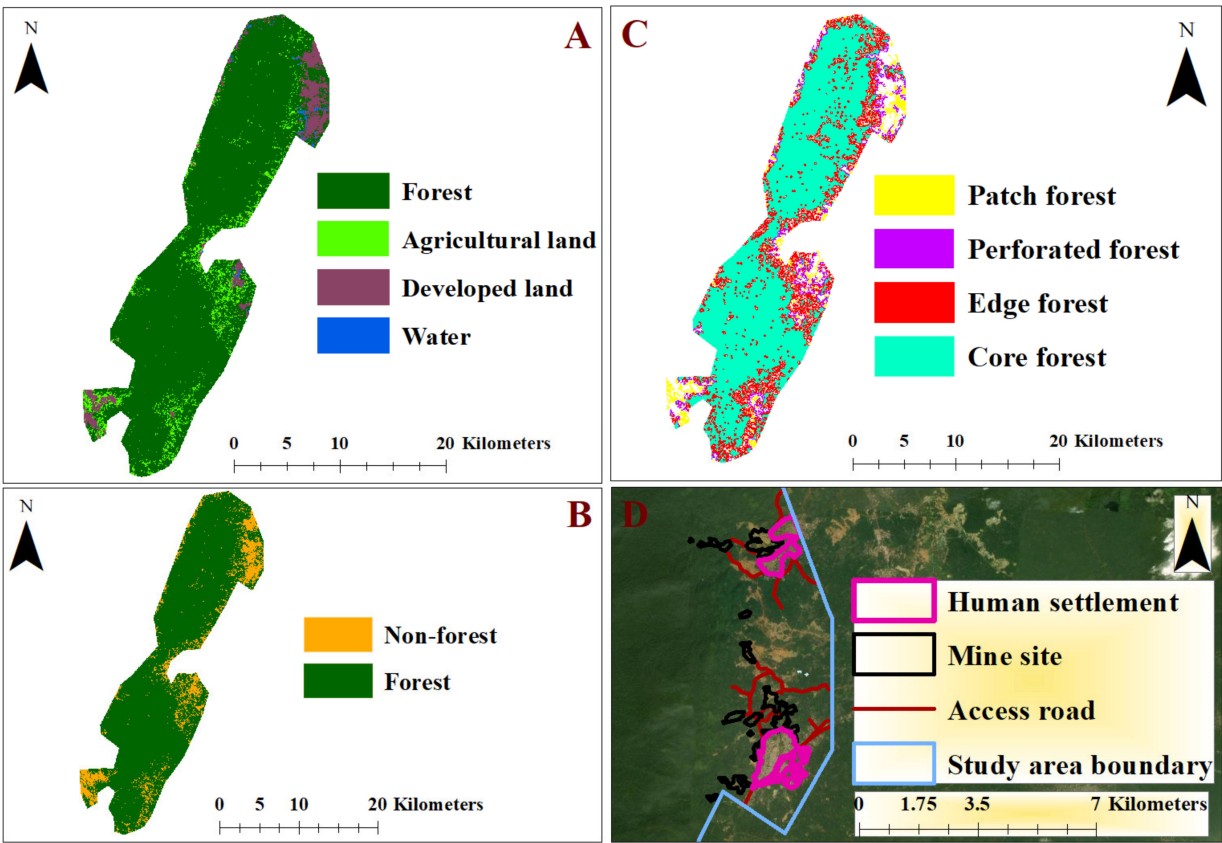

**Figure 2.** Classified Landsat image showing land categories and forest/non-forest patches. (**A**) shows the land classes, (**B**) shows the forest/non-forest patches, (**C**) shows the visual impression of the morphological analysis output indicating the categories of forest fragmentation and (**D**) shows locations of some of the land use footprints in the Atewa Range Forest Reserve.

In this study, we randomly selected 125, 108, 95, and 69 training samples from the Landsat image, respectively for forest cover, agricultural, developed land, and water to train the RF classifier. The final land classes, noted above, were assessed for accuracy using 550 ground-truth data sampled from the ARFR using a global positioning system (GPSMap 78s). With reference to the ground truth data, the accuracy of the classified image has been detailed in Appendix A.

The classified Landsat image was further categorized into forest and non-forest as a requirement for landscape morphological analysis [18] (see Figure 2, Panel B). The landscape fragmentation tool (LFT), an ArcGIS extension, was used to perform the landscape morphological analysis to classify the landscape into patch forests, edge forests, perforated forests, and core forests to be able to select the core forests for further analysis (see Table 1 for detailed descriptions; also, see Figure 2, Panel C). During landscape morphological analysis, land cover maps in binary raster format are classified into spatial patterns at per-pixel level [18]. "An algorithm to classify forest patterns is defined by a sequence of logical operations such as union, intersection, complementation, and translation using geometric objects called 'structuring elements' (SE) of pre-defined shape and size" ([18], p. 172). Thus, it is important to note that the LFT algorithm for forest pattern analysis uses the originally classified forest and non-forest patches to construct new forest patches using a variety of operations, including but not limited to combining and intersecting some of the patches, to produce a model of forest landscape attributes. Previous studies have used the LFT to perform morphological analysis to determine landscape integrity in both urban wetlands and rural landscapes [38,39]. With the LFT, we identified 856 patch (degraded) forests,

530 edge forests, 35 perforated forests, and 516 core forest patches. The minimum and maximum core forest patches were 0.058 ha and 21,823 ha, respectively.

**Table 1.** Categories of forests and their descriptions.

| Categories | Description |
|---|---|
| Patch forests | Patch forests are small fragments of forest that do not contain any core forest pixel, and these are degraded by 'edge effect' [a] |
| Edge forests | Edge forests are fragmented forests found in the edge effect area, and they are found along the edges of non-patch tracks. |
| Perforated forests | perforated forests are fragmented forests that occur in the edge effect and along small clearings in the non-patch tracks. |
| Core forests | Core forests occur outside the edge effect area. These are intact forests and are not degraded by fragmentation |

[a] A distance of 100 m from non-forest to the forest interior where human activities, wildfires, micro-climate effects, and the effects of fragmentation would likely degrade forest cover [16,17]. Note: the definitions of the categories of forest fragmentation are based on definitions from the Center for Land Use Education and Research [19] and Vogt et al. [18].

The spatial extents of land use footprints (mining activities, access roads, human settlements, and logging sites) were digitized from the Google Earth Pro high-resolution aerial photo (see, e.g., Figure 2, Panel D). The land uses were digitized at scales ranging between 1:5000 and 1:15,000. Additionally, from the DEM, we calculated the elevation (measured in meters), slope (measured in degrees), and aspect (direction of the slope measured in degrees) of the landscape where patches of forest are found. The calculations of the slope and aspect were done in ArcGIS 10. In ArcGIS, we calculated the distances between the patches of forest cover identified by the LFT. We indicated the patches of forest cover (patch forests, perforated forests, edge forests, and core forests) that are within and beyond 1 km from land uses based on Haddad et al. [40] who showed that most of the remaining forest fragments in the world are within 1 km of major land uses (agriculture, logging, human settlement, etc.). Such a criterion has been used in previous studies that measure how patches of different sizes are arranged in connection with land uses [30]. Thus, this criterion is consistent with a previous study. Here, in the current study, we hypothesize that intact patches (core forest) are more likely to be found beyond 1 km of land use. Thus, from this hypothesis, we categorized core forests to be either within or beyond 1 km from the land uses. Whereas within and beyond 1 km thresholds were used to categorize the spatial location of land uses in relation to core forests, it is important to note that both the land uses and core forest patches are found in different locations as shown in Figure 2.

### 2.4. Variables and Analytical Framework

Since the phenomenon of interest is the core forest, after the landscape morphological analysis, we selected and labeled patches as core forest = 1 and non-core forest patches = 0. Thus, the probability of patches being core forest is 1 or otherwise, 0 is selected as the dependent variable. Distance to mining sites, distance to logging sites, distance to agricultural land, distance to access roads, distance to human settlements, and terrain variables (elevation, slope, and aspect) were the independent variables used in the analysis. Distance to agricultural land, distance to logging sites, distance to mine sites, distance to access roads, and distance to human settlements were treated as categorical variables, with the indication of which forest patches are beyond 1 km from these land uses or otherwise. For the terrain variables, aspect was treated as a categorical variable, but elevation and slope were considered continuous variables. Based on the Gujarati and Porter [41] framework for building a model, we developed a spatially explicit logistic regression model to ex-

plain the occurrence of core forest patches relative to land uses and terrain characteristics (see Equation (1)).

$$\text{CFP} = \left(\frac{P_1}{1 - P_1}\right) = \beta_0 + \beta_1 EL + \beta_2 SL + \beta_3 AS + \beta_4 MS + \beta_5 AL + \beta_6 AR + \beta_7 LS + \beta_8 HS + e \quad (1)$$

where $P_1$ is the probability that core forest patches (CFP) are occurring in the forest landscape [CFP is 1, otherwise 0]; $\beta_0$, $\beta_1$, ..., $\beta_8$ are the model's coefficient [Note: $\beta_0$ is a constant]; *EL* is elevation [in meters]; *SL* is slope [in degrees]; *AS* represents aspect; *MS* is the designated symbol for a distance beyond 1 km to mine sites [Yes = 1, No = 0]; *AL* represents a distance beyond 1 km from agricultural land uses [Yes = 1, No = 0]; *AR* represents a distance beyond 1 km from access roads [Yes = 1, No = 0]; *LS* represents a distance beyond 1 km from logging sites [Yes = 1, No = 0]; and *HS* represents a distance beyond 1 km from human settlements [Yes = 1, No = 0]; and *e* is an error term. Land use and terrain variables have been used in previous studies to model the forest patch sizes and the occurrence of vegetation cover [30,42]. Thus, our variable selection in this study is grounded in successful empirical studies from other tropical forest regions. We performed Pearson's Correlation analysis to show the strength of relationships between the dependent variable and the independent variables.

We used the receiver operating characteristics (ROC) curve statistic to determine the robustness of the model. The ROC statistic shows the likelihood of either having a true positive (i.e., predicting correctly) or a false positive (i.e., predicting falsely) [43]. In dichotomous statistical modeling (e.g., binary logistic regression), the ROC statistic provides a very useful way of determining the predictive power of the selected model. The predicted likelihoods are a continuous indicator to be compared to the observed binary response variable [43]. This ROC statistical approach imposes a threshold of greater than 0.5 with a perfect score of 1 showing a perfect classification [44]. Moreover, we test the robustness of the model using the omnibus test of model coefficient (OTMC). This test determines whether a final model (model with explanatory variables included) is improved as compared to the base model (model without explanatory variables). For the OTMC, the final model is significant at a *p*-value of 0.05 or less.

## 3. Results

### 3.1. Summary of Categorical Variables and the Land Use-Distance Threshold

Our correlation analysis results show that there is a statistically significant relationship between core forests and our independent variables such as elevation, slope, aspect, human settlement, logging sites, mine sites, agricultural sites, and access roads (see Table A2). The outcome of the analysis reveals that most of the patches are on the east-facing, southeast-facing, and northwest-facing slopes of the landscape (see Appendix A Table A3). The outcome of the study reveals that most of the forest patches on the landscape are found beyond the 1 km land use-distance threshold. For instance, 1099 out of 1937 forest patches are found beyond 1 km from agricultural land uses, that is, the footprints that come from the main economic activity of the communities around the ARFR landscape (see Appendix A Table A4; also, refer to Figure 2 Panel A for the distribution of agricultural footprints in the ARFR).

### 3.2. Core Forest Patches and Their Relationship with Land Use and Terrain Variables

From the analysis, we show that a unit reduction in elevation would significantly likely reduce the possibility of forest patches being core forests on the landscape by 0.995 times (approximately 0.5%). On the other hand, a unit increase in slope gradient significantly increases the odds of a forest patch being among the core forest category by 1.35 times (approximately 35%) (Table 2). Even though most of the forest patches are found on the east-facing slope of the landscape, the logistic regression model shows it is 1.70 times significantly more likely for the forests to be core forest patches on the northeast-facing side of the slope as compared to the east-facing side. That is, the odds of the forest patches being

the core forest increase by approximately 70% on the northeast-facing slope as compared to the east-facing slope (Table 2).

**Table 2.** Land use and terrain variables associated with the occurrence of core forest patches.

| Variables | B | S.E. | Wald | Df | *p*-Value | AOR | 95% C.I. for AOR Lower | Upper |
|---|---|---|---|---|---|---|---|---|
| Elevation | −0.005 | 0.001 | 38.533 | 1 | 0.000 ** | 0.995 | 0.994 | 0.997 |
| Slope | 0.299 | 0.037 | 64.343 | 1 | 0.000 ** | 1.349 | 1.254 | 1.451 |
| Aspect (East) | | | 116.663 | 7 | 0.000 ** | | | |
| Aspect (North) | −0.231 | 0.404 | 0.328 | 1 | 0.567 | 0.793 | 0.360 | 1.751 |
| Aspect (Northeast) | 0.527 | 0.203 | 6.751 | 1 | 0.009 ** | 1.694 | 1.138 | 2.521 |
| Aspect (Northwest) | -1.214 | 0.188 | 41.513 | 1 | 0.000 ** | 0.297 | 0.205 | 0.430 |
| Aspect (South) | −0.150 | 0.251 | 0.358 | 1 | 0.549 | 0.861 | 0.526 | 1.407 |
| Aspect (Southeast) | −0.379 | 0.163 | 5.408 | 1 | 0.020 ** | 0.684 | 0.497 | 0.942 |
| Aspect (Southwest) | −0.472 | 0.231 | 4.183 | 1 | 0.041 ** | 0.624 | 0.397 | 0.981 |
| Aspect (West) | −1.951 | 0.237 | 67.605 | 1 | 0.000 ** | 0.142 | 0.089 | 0.226 |
| Beyond 1 km from human settlements (No) | | | | | | 1 (ref) | | |
| Beyond 1 km from human settlement (Yes) | 0.251 | 0.155 | 2.613 | 1 | 0.106 | 1.285 | 0.948 | 1.742 |
| Beyond 1 km from agricultural land (No) | | | | | | 1 (ref) | | |
| Beyond 1 km from agricultural land (Yes) | 0.155 | 0.113 | 1.882 | 1 | 0.170 | 1.167 | 0.936 | 1.456 |
| Beyond 1 km from logging sites (No) | | | | | | 1 (ref) | | |
| Beyond 1 km from logging sites (Yes) | 0.470 | 0.162 | 8.407 | 1 | 0.004 ** | 1.600 | 1.164 | 2.198 |
| Beyond 1 km from mine sites (No) | | | | | | 1 (ref) | | |
| Beyond 1 km from mine site (Yes) | 0.331 | 0.133 | 6.182 | 1 | 0.013 ** | 1.392 | 1.073 | 1.807 |
| Beyond 1 km from AR (No) | | | | | | 1 (ref) | | |
| Beyond 1 km from AR (Yes) | 0.761 | 0.162 | 22.113 | 1 | 0.000 ** | 2.141 | 1.559 | 2.941 |
| Constant | −1.158 | 0.265 | 19.082 | 1 | 0.000 ** | 0.314 | | |

Note: ** indicates a statistically significant relationship between a specific independent variable and the occurrence of core forest patches. SE is the standard error, df is the degree(s) of freedom, AOR is the adjusted odds ratio(s), and CI is confidence intervals.

The study finds that it is 1.60 (approximately 60%) times significantly more likely for forest patches to be core forest patches if they are found beyond 1 km from logging sites. The model coefficient also indicates that as distance increases from the logging sites, there is a significant possibility that core forest patches would occur (Table 2). Similarly, we show that mine sites are associated with the occurrence of core forest patches. The study finds that it is 1.40 (approximately 40%) times significantly more likely for forest patches to be core forest patches if they are beyond 1 km from the mine sites. Thus, the model coefficient further indicates that there is a positive relationship between distances from mine sites and the occurrence of core forest patches (Table 2).

Moreover, the study finds that access roads are significantly associated with the occurrence of core forest patches. Specifically, the outcome of our analysis shows it is 2.14 times significantly more likely for forest patches to be core forest patches if they are beyond 1 km from access roads. Thus, this shows that the odds of core forest patches occurring increase by 114% if patches are beyond 1 km from the access roads. The coefficient of the model indicates that as the distance between access roads and forest patches increases, there is a significant possibility that core forest patches would occur (Table 2).

### 3.3. Model Robustness

Our analysis shows two different levels of model robustness with different measures. First, the study finds the area under the curve (Figure 3) ROC statistic of 0.775. Second, the outcome of the OTMC indicates a significant difference (improvement) between the base model (i.e., the model with no explanatory variables) and the final model (i.e., the model with all explanatory variables) (see Table 3).

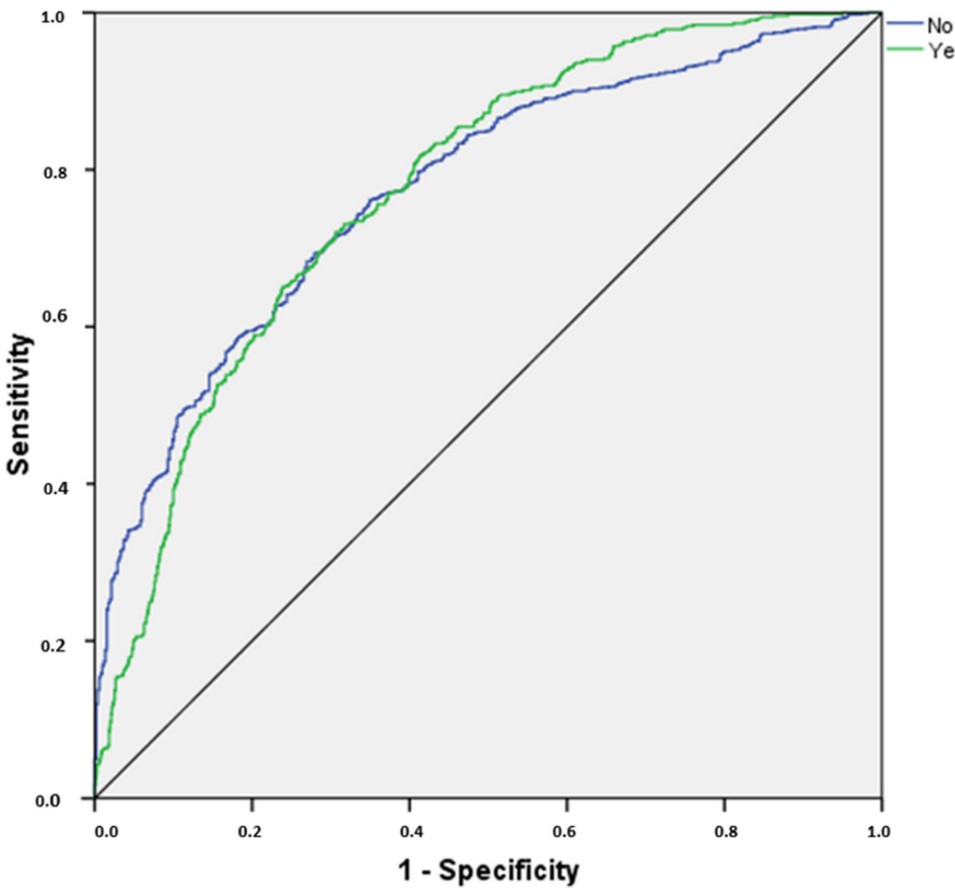

**Figure 3.** Area under the curve showing the occurrence of core forest patches as a classified binary ariable.

**Table 3.** Omnibus Tests of Model Coefficients.

|  | Chi-Square | Df | Sig. |
|---|---|---|---|
| Step | 252.534 | 14 | 0.000 ** |
| Block | 252.534 | 14 | 0.000 ** |
| Model | 252.534 | 14 | 0.000 ** |

Note: ** for the Omnibus Tests of Model Coefficients, the final model is significant when the *p*-value is 0.05 or less.

## 4. Discussion

We present a model showing the relationship between the occurrence of core forest patches and land use and terrain variables. A significant association between the occurrence of core forest patches and elevation, slope, aspect, logging sites, mine sites, and access roads has been identified through the modeling process. With this association being established, the ROC value of 0.775 achieved in this analysis suggests that a modest, acceptable, and robust model has been produced. Similar ROC values (e.g., 0.715 and 0.790) have been reported in a previous study that used terrain and distance-land use variables in analyzing forest cover conversion patterns [45,46]. Furthermore, from our analysis, the outcome of the OTMC suggests that the final model results we have presented are a significant improvement upon the base model, the model without any explanatory variables.

Whereas our study compares results with that of previous studies from other tropical forest regions, studies that directly compare to ours are rare, and thus we make comparisons with studies that mostly focused on general land use and forest cover change. Our study outcome suggests that on higher elevations and higher slopes, intact forests are more likely to occur. This study outcome is likely due to the lack of easy accessibility at the higher

slope and elevations by humans, and thus human activities would less likely take place on higher slopes and elevations. Also, the results of the study suggest that the high slopes and altitudes should likely be an area of focus in terms of crafting land management strategies and policies to protect the existing intact forest patches. With this being mentioned, this study outcome also creates an opportunity for the establishment of land policies that would likely improve sustainable land management in low-lying areas and low slopes on forest landscapes. Our results corroborate a related study in Chunati Wildlife Sanctuary, Bangladesh, in which Islam et al. [42] noted that an increase in elevation and slope increases the probability that a vegetation patch will occur. However, Islam et al. [46], did not indicate whether or not these vegetation patches are the core type. Concerning the direction of slope, Yang et al. [47] have shown that north- and south-facing slopes influence differences in vegetation structure (biomass, cover, and height) in southwest China. In other tropical regions of the world (e.g., Indonesia and India), such terrain variables play roles in the occurrence of vegetation patches [48,49]. Thus, this demonstrates that terrain variables influence the nature of vegetation cover in other locations similar to what we have found in our study. However, the differences, for instance, in how the direction of slope variable influences vegetation structure may also be determined by latitudinal differences and the amount of solar radiation received in the various environments [50,51]. These differences present unique environmental challenges in different landscapes and thus these challenges would require unique land management strategies.

The study results note that logging in the forest reserve is significantly associated with the occurrence of core forests. Specifically, our results suggest that areas close to logging sites would likely not have more of the core forests as compared to areas far from the logging sites, and thus, intact forests would likely be located far from the logging sites. The outcome of our study is related to the results of a previous study in the Brazilian Amazon which found that logging activities give rise to non-contiguous forests in nearby areas culminating in the formation of many smaller fragments of forest cover [52]. Similarly, in a related study that covers Africa, Asia, and Latin America, Putz et al. [53] noted that the percentage of intact forests declines within logging blocks as the intensity of harvest increases. The rate at which non-core forests develop would increase at an increasing rate as more logging activities continue to penetrate core forest areas [52]. Thus, with logging activities threatening most tropical forest regions, relevant land management measures would be necessary, especially in locations where logging activities have become a greater component of people's livelihoods.

Similar to the logging activities, mining activities in the forest reserve have grown in recent times, and our study results show a significant association between mine sites and the occurrence of core forest patches. The study results suggest that core forests would more likely occur in areas far away from the mine sites. This outcome was expected because of the rate at which mining activities have increased in recent times in the forest reserve. Recent studies have noted that mining activities in tropical regions (e.g., India, Ghana, Colombia, Venezuela, Guyana, Suriname) contribute to a significant decline in forest cover [22,54–57], and these are likely to affect the core forest patches. However, the use of sites in the forest landscape for mining would likely open up areas near the sites for other activities (e.g., the creation of mining infrastructure), and these are likely to result in further fragmentation and thus, the occurrence of fragmented forests near the mining sites. With our results, land managers could create land policies that would ensure core forest patches are protected by controlling the spatial spread of mine sites to many parts of the protected landscape. Thus, the results imply that land rehabilitation plans and policies should target the areas near the mine sites while increasing protection regimes in the areas farther away from these mine sites.

The results of our analysis suggest that the occurrence of core forest patches has a significant association with the presence of access roads in the forest reserve. Specifically, the study outcome suggests that core forest patches would likely be found in areas farther away from the access roads as compared to the areas closer to the roads. The areas close to

the road would likely be accessed by humans, and thus, forests in those areas would likely be degraded because these areas would be opened to different levels of human influences. Hence, intact or core forest patches are less likely to be found close to the roads. Direct measurements of the impacts of access roads have shown that road construction has led to deforestation [16]. As noted by Laurance et al. [16], the degradation of forests contributed by roads in tropical forests is growing rapidly and is in two forms. First, through the clearance of forest for construction and second, through opening up of forest areas for humans to access hitherto inaccessible areas and thus, increasing the likelihood of clearing accessible forest areas for other human activities. Moreover, previous study results corroborate our results in that analysis from major tropical forest areas in Africa, Asia, and Latin America has found that intact forests are mostly found in areas far from access roads [53]. In a related study, Teixeira et al. [58] in an analysis of the Plateau of Ibiúna, southeast Brazil noted that forest regeneration significantly occurs in areas farther from roads. Thus, intact forests would likely form in these areas far from the road. The relationship between various land uses and the occurrence of core forest patches as indicated in the results calls for management measures and policies that protect forest patches that are farther away from land uses. Additionally, for managing the already fragmented forests, this study outcome shows that forest patches close to the land uses are mostly non-core forests and thus, management initiatives (e.g., tree planting) meant for regrowing and creating intact forest patches should target the areas close to land uses.

The results of our analysis show that forest patches are more likely to be core forests if they are found beyond 1 km from human settlement and agricultural land. However, the association between these human activity footprints and core forests is not statistically significant, implying the occurrence of core forest patches within or beyond 1 km from the two land uses is by random chance. Upon reviewing the spatial locations of human settlements, it was noticed that forests in general are not found in areas where human settlements are located, and it is likely that is the reason the association is not statistically significant. On the other hand, agricultural footprints are not associated with core forest patches likely because as found in a previous study [59] agricultural lands are more likely to transition to forest cover in the future, and thus, the agricultural land locations would likely not be a problem and that would likely be the reason why they are not associated with whether or not core forest would occur. However, it is important to note that there will be a need to test more hypotheses to establish a robust conclusion about human settlements and agricultural land footprints and their association with core forest patches.

## 5. Conclusions

This study modeled the relationship between the occurrence of core forest patches and dominant anthropogenic activities and terrain variables. We show that core forest patches are significantly associated with land use and terrain variables. With a robust model, we conclude that core forest patches are found far away from land uses. Similarly, on higher slopes and elevations, core forest patches are more likely to occur relative to lower slopes and elevations. Moreover, our findings, based on the direction of slope, indicate that land managers have unique tasks of monitoring the different directions of slopes and ensuring that there is sustainable management of forest patches in protected landscapes. Thus, land management and land policies for maintaining core forests should target areas close to land uses as well as low slopes and elevations. Conversely, core forest preservation policies could target areas far away from land uses and areas on higher slopes and elevations.

With the current state-of-the-art geospatial satellite data processing having limitations, the classification of forest patches is less likely to be perfect given that there may be forest patch sizes that are less than the spatial resolution of the Landsat image used in this study. Thus, such a limitation would likely underestimate the number of patches and consequently the underestimation of the patches for the landscape morphological processing. Additionally, the selection of within 1 km and beyond 1 km distance threshold to classify the occurrence of core forest would likely be a limitation in this study. The lim-

itations would slightly systematically bias the results. Hence, we recommend the use of high-resolution satellite images for classifying the forest patches as well as using multiple distance thresholds to classify the occurrence of forest patches in future studies.

**Author Contributions:** Conceptualization, J.O.A. and W.A.-D.; methodology, J.O.A.; software, J.O.A.; validation, W.A.-D., D.A. and J.O.A.; formal analysis, J.O.A. and W.A.-D.; investigation, J.O.A., D.A. and W.A.-D.; resources, J.O.A.; data curation, W.A.-D. and J.O.A.; writing—original draft preparation, J.O.A., D.A. and W.A.-D.; writing—review and editing, J.O.A., D.A. and W.A.-D.; visualization, J.O.A., D.A. and W.A.-D.; supervision, J.O.A., D.A. and W.A.-D.; project administration, J.O.A., D.A. and W.A.-D. All authors have read and agreed to the published version of the manuscript.

**Funding:** This research received no external funding.

**Data Availability Statement:** Data processed and analyzed in this study are from publicly available USGS archive (https://earthexplorer.usgs.gov/).

**Conflicts of Interest:** The authors declare no conflict of interest.

## Appendix A

**Table A1.** Results of the accuracy assessment of classified Landsat image from the Atewa Range Forest Reserve.

| | | Ground Truth Samples | | | | | | |
|---|---|---|---|---|---|---|---|---|
| | **Category of Land** | **Forest Cover** | **Agricultural Land** | **Developed Land** | **Water** | **Total Truths** | **User Accuracy (%)** | **Commission Error (%)** |
| Predicted land class | Forest cover | 180 | 5 | 1 | 0 | 186 | 96.77 | 3.23 |
| | Agricultural land | 5 | 128 | 1 | 1 | 135 | 94.81 | 5.19 |
| | Developed land | 2 | 3 | 122 | 1 | 128 | 95.31 | 4.69 |
| | Water | 1 | 0 | 0 | 100 | 101 | 99.01 | 0.99 |
| | Total | 188 | 136 | 124 | 102 | 550 | | |
| | Producer accuracy (%) | 95.74 | 94.12 | 98.39 | 98.04 | | | |
| | Omission error (%) | 4.26 | 5.88 | 1.61 | 1.96 | | | |
| | Overall accuracy (%) | 96.36 | | | | | | |

**Table A2.** Results of Pearson's correlation analysis.

| | **1** | **2** | **3** | **4** | **5** | **6** | **7** | **8** | **9** |
|---|---|---|---|---|---|---|---|---|---|
| 1. Core Forest | 1 | | | | | | | * | |
| 2. Elevation (meters) | 0.061 ** | 1 | | | | | | | |
| 3. Slope | 0.186 ** | 0.382 ** | 1 | | | | | | |
| 4. Aspect | −0.193 ** | −0.181 ** | 0.095 ** | 1 | | | | | |
| 5. Human settlement | 0.127 ** | 0.465 ** | 0.358 ** | −0.019 | 1 | | | | |
| 6. Logging site | 0.069 ** | 0.337 ** | 0.098 ** | −0.132 ** | −0.036 | 1 | | | |
| 7. Mine site | 0.124 ** | 0.452 ** | 0.280 ** | −0.040 | 0.328 ** | 0.171 ** | 1 | | |
| 8. Agricultural land | −0.064 ** | −0.180 ** | −0.101 ** | 0.115 ** | −0.066 ** | −0.163 ** | −0.074 ** | 1 | |
| 9. Acess roads | 0.149 ** | 0.563 ** | 0.364 ** | 0.039 | 0.600 ** | 0.103 ** | 0.450 ** | −0.032 | 1 |

$* \ p < 0.05$, $** \ p < 0.01$.

**Table A3.** Patches of forest cover and the direction of slope variable.

| Occurrence of Patches | | Aspect (Direction of Slope) | | | | | | | | |
|---|---|---|---|---|---|---|---|---|---|---|
| | | **East** | **North** | **Northeast** | **Northwest** | **South** | **Southeast** | **Southwest** | **West** | **Total** |
| **CF** | No | 346 | 18 | 94 | 246 | 91 | 257 | 114 | 255 | 1421 |
| | Yes | 181 | 13 | 72 | 65 | 29 | 93 | 36 | 27 | 516 |

Note: CF represents core forest.

**Table A4.** Patches of forest cover and the distance-land use variables.

| Occurrence of Patches | | Beyond 1 km from Human Settlement | | Total |
| --- | --- | --- | --- | --- |
| | | No | Yes | |
| Core Forest | No | 571 | 850 | 1421 |
| | Yes | 136 | 380 | 516 |
| | | Beyond 1 km from agricultural land | | Total |
| | | No | Yes | |
| Core Forest | No | 619 | 802 | 1421 |
| | Yes | 219 | 297 | 516 |
| | | Beyond 1 km from logging sites | | Total |
| | | No | Yes | |
| Core Forest | No | 325 | 1096 | 1421 |
| | Yes | 85 | 431 | 516 |
| | | Beyond 1 km from mine sites | | Total |
| | | No | Yes | |
| Core Forest | No | 619 | 802 | 1421 |
| | Yes | 154 | 362 | 516 |
| | | Beyond 1 km from access roads | | Total |
| | | No | Yes | |
| Core Forest | No | 754 | 667 | 1421 |
| | Yes | 187 | 329 | 516 |

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
