# Peer review of "Land Use and Landscape Characteristics Are Associated with Core Forest Patches in Ghana"

_land, doi:10.3390/land12010071_

Round 1
Reviewer 1 Report
2022-11-21
Review of the manuscript entitled “Spatial distribution of land uses and terrain characteristics are associated with core forest patches in the Atewa Range Forest Reserve in Ghana” submitted to the editorial board of the Land journal.
General comments
Forest fragmentation is a very complex phenomenon, which consists primarily in the transformation of large and continuous forest areas into a larger number of smaller and isolated patches. It is considered a serious threat to biodiversity. It leads not only to the loss of natural habitats, but also to the launch of often negative processes that modify the structure of forest patches. Therefore, the article submitted for review concerns an important practical issue of protection of fragmented forest communities under the influence of social and economic factors.
Studies on forest fragmentation is quite common. They include both the analysis of the effects and causes of fragmentation at different spatial scales. For this purpose, a wide range of methodological and technological approaches have been developed and applied. Therefore, the general knowledge about forest fragmentation is quite broad, but we know less about the impact of local natural and socio-economic conditions on the features of fragmented forest ecosystems.
The work submitted for evaluation models the relationship between the spatial distribution of the main forest patches, land use and terrain variables. The Authors used a wide range of data and analysis methods for this purpose. Indeed, they used both geospatial machine learning techniques and statistical methods to process satellite imagery and to model the relationship between major forest patches and related variables. Nevertheless, the obtained results do not contribute much to the general state of knowledge on forest fragmentation, mainly because in the methodological part the authors did not fully explain the adopted methodological solutions. Therefore, they should be supplemented and clarified.
Detailed notes
• The study was undoubtedly of a local nature, therefore it is unfortunate that the arbitrary (or based on the literature) adopted a global indicator, i.e. the distance (1 km) of core forests from the main areas of land use. Especially since some core forests were located within 100m of non-forests.
• Why was the actual distance between the core forests and the different types of land use in the study site not specified? The paper also does not present the results of area assessment taking into account the category of forest patches and other areas of land use. In the opinion of the reviewer, this reduced the cognitive value of the work, which does not present an analysis of the impact of local conditions on the degradation and fragmentation of forests
• • Many assumptions were introduced into the research method, which limit the possibility of proper interpretation of the obtained results. This should be considered the use of the same distance of 1 km to core forests from different types of use. This means that fragmentation and degradation of forests are equally affected by different forms of land use
• Only core forests were used for the study? For what purpose have forests been divided into categories?
• Has a minimum area of core forests been established?
• Were the relationships between the variables used in the model investigated (for example, using a correlation coefficient?)
• The discussion is practically focused on the analysis of the distance of core forests from other areas of use However, it is worth using the term "accessibility of core forests” more often in the work, which is influenced not only by the distance, but also by roads and terrain.
Despite these shortcomings, however, the effort of the authors of the article should be appreciated for the effort put into the preparation of the study, which after introducing corrections is worth publishing.

Reviewer 2 Report
This paper provides an interesting and innovative study by using machine learning geospatial techniques and statistical methods to process satellite images and model the relationship between the spatial distribution of core forest patches, land uses, and terrain variables in the Atewa Range Forest Reserve in Ghana. However, I recommend several areas that should be addressed and clarified:
1. It is necessary and useful to provide the meaning of “core forest patch” in the introduction section when the “core forest patch” first appears.
2. I would suggest authors further explain why takes the case of Atewa Range forest reserve to model the relationship between the spatial distribution of core forest patches, land uses, and terrain variables, for instance, its representativeness as one of the major tropical forest areas. Such an explanation may enable to provide more support for the discussion section, for instance, to compare with previous studies of major forest areas.
3. Similarly, it is suggested to provide information and evaluation about existing policies and programs of the study area, which would be useful for developing the discussion section and then summarizing the research findings and possible coping strategies.
4. Authors should further explain why study the relationship between the spatial distribution of core forest patches, land uses, and terrain variables. As lines, 62-63 showing, “However, measuring the relationship between core forest patches, land uses, and other associated factors were beyond the scope of the previous studies”, thus, besides land uses and terrain variables, whether there are other (potential) associated factors or not? If yes, why land uses and terrain variables are focused on? Similarly, authors should further explain when talking about land uses and terrain variables, why these variables, including EL, SL, MS, AL, AR, LS, HS, etc., are selected.
5. The discussion and conclusion sections should be strengthened. Some key findings should be further explained and clarified.
(1) “The study finds that the odds of forest patches being core forests significantly increase by 1.60 and 2.14 if patches are found beyond 1 km from logging sites and access roads, respectively” (see lines 15-17, and lines 243-244). Although the authors explain the reasons for such a criterion of 1 km, I am curious whether it is possible to provide an upper limit or classify the distance length (beyond and within 1 km) into several categories. Based on this, a deeper quantitative analysis could be developed.
(2) To some extent, some conclusions including coping strategies are those scholars, governments, and even the public already reached or easily reach an agreement. For example, “intact forest patches would likely be found far away from land uses. Thus, the protection of forest patches should target areas far from land uses whereas forest restoration programs should target areas close to land uses” (see lines 17-19). I would suggest authors further summarize the research findings (particularly some quantitative findings), and connect and compare them with the existing studies, rather than merely illustrating that the research outcomes are related to and/or corroborate some existing studies. Please pay attention to the difference when talking about existing studies in the literature section and in the conclusion section. Moreover, it is necessary and significant to provide possible implications and more explicit coping strategies in the discussion and conclusion sections.
6. Some errors should be revised. For example, in line 331 half a bracket is missing.
Reviewer 3 Report
Review land-2063521
The present paper entitled “Spatial distribution of land uses and terrain characteristics are associated with core forest patches in the Atewa Range Forest Reserve in Ghana” aimed to identify the main factors determining the relationship between core forest patches, land use type and landscape characteristics. A GIS approach was used to process land use and terrain characteristic data necessary. A logistic regression model was used to disentangle the relationship between core forest patch and different land use types and terrain characteristics. The outcome of this study showed that core forest patches are significantly related to the distances to mining sites and access roads and aspect. Interestingly, distance to human settlements and agricultural land use has no effect. This is a nice study and suitable for publication in land.
However, from an ecological view, this study provides not is not very exciting new results. Furthermore, I have several questions concerning the definition used and the kind of data analysis.
Listed below are some major and minor comments for the authors, which I have broken down into the main section of the manuscript:
Title: very long: e.g. Core forest patches are associated with land use and landscape characteristics.
Abstract: OK
Introduction:
Lines 35–37, 62 and throughout the manuscript: The definitions used for core forest respectively core forest patches are missing in the manuscript. What about the shape of the forests? All forests differed in their shape. Even large forest with a complex shape can have only a small core forest area. Furthermore, the core area is defined "as the area within patch beyond some specified depth of edge influence (i.e., edge distance) or buffer width". All these definition are missing in the present version of the manuscript.
Material and Methods
Line 123: An important factor can be the intensity of land use; e.g. agricultural land use intensity
Lines 142–143 and Table 1: From an ecological view, the used definitions are not understandable and not releated to the ecological term used.
Lines 156–157: Data of intensity of the different land use types?
Lines 163–170 and Lines 188–190: Criteria for choosing this 1 km distance? Why not data of the real distances between forest core patches and the different land use types? This kind of data can be used as continuous variables in the model.
Lines 188–190: Are the used land use types inter-correlated?
Results:
Table 2 and Table 3 can be moved to the supplementary material
Table 4: heading: change core patches to core forest patches
Discussion: OK
Reviewer 4 Report
This study built a logistic regression model to identify the relations between the core forest appearance and land uses as well as terrain characteristics. Although the method is kind of simple, the results are valuable. There are a couple of points where it could be stronger, the comments are as follows:
In Abstract section,
- Please summarize all the findings in the abstract. Only the land use results were available at present, so please supplement the terrain-related results.
- If possible, please write a short sentence to describe your purpose and significance of this research at the end of this part.
In Keywords section,
- There were too many keywords, please try to reduce to 3-4 if possible, and keep each keyword within three words
In Introduction section,
- This part only demonstrated the land uses induced the forest cover degradation, however, the terrains influences were not mentioned, please supplement them.
- L56-157, the statement was not clear, please rewrite it.
In Materials and methods section,
- How big is the reserve? Please state the area of ARFR in part 2.1.
- Fig.1, the maps should be standardized, as the color of A, B and C, the unit of scale bars, and the rectangle size in legend should be unified, also I suggest changing a lighter color for the north arrow, because now it is kind of blurred. Moreover, Line 99, panel B shows……
- I think the organization was disorder in 2.3. The classification method should be mentioned first, and then explained why RF was selected, and illustrated the classification results and accuracy at last.
- Why not pansharpened the image first and then classified the land use types? The spatial resolution was relatively low for the classification in a reserve, which you have mentioned in conclusions part, so why not use the band 8 to improve the resolution?
In Results section,
- In section 3.2, Table 4 should be concluded in the first paragraph, and the last paragraph which showed the model robustness should be removed to a separate section as 3.3.
In Discussion section,
- This part is OK, but I recommend adding some explanations on why other variables such as human settlements and agricultural land were not significantly associated with the occurrence of core forest, cause such statements would strengthen the results.
- In addition, there were too many sentences began with “This study outcome” or” The study outcome”, which made the readers feel tedious and boring.
Round 2
Reviewer 2 Report
The authors addressed some comments that I proposed in the last round, yet there are still some areas that should be addressed and clarified in my point of view:
1. In the introduction section, lines 45-47 when defining “core forests” show repeated information. As the authors wrote, “Core forests occur outside the edge effect area. These are intact forests and are not degraded by fragmentation”, however, what does “edge effect area” mean? I tend to believe it is necessary to clarify the definition of “edge effect area” and where the definition of “core forests” is from? Was it defined by the authors or other scholars?
2. Some descriptions provide repeated information, e.g., lines 60-63, and lines 107-109.
3. As I mentioned in the last round, the possible policy implication is suggested to be strengthened in the discussion and conclusion sections.
4. The paper made some errors, including spelling mistake, e.g., “infleunce”(see line 74), “definitiions” (line 246), and incomplete sentences, e.g., lines 106-107.
Author Response
Reviewer 2 Comments and Suggestions for Authors
The authors addressed some comments that I proposed in the last round, yet there are still some areas that should be addressed and clarified in my point of view:
- In the introduction section, lines 45-47 when defining “core forests” show repeated information. As the authors wrote, “Core forests occur outside the edge effect area. These are intact forests and are not degraded by fragmentation”, however, what does “edge effect area” mean? I tend to believe it is necessary to clarify the definition of “edge effect area” and where the definition of “core forests” is from? Was it defined by the authors or other scholars?
Our Response: Authors have removed the repeated definition. Also, authors have provided the definition of edge effect area. We have provided citations for all definitions [core forest, edge effect zone]
- Some descriptions provide repeated information, e.g., lines 60-63, and lines 107-109.
Our Response: Authors have removed the repetitions from lines 60-63 and 107-109 and revised the affected sentences.
- As I mentioned in the last round, the possible policy implication is suggested to be strengthened in the discussion and conclusion sections.
Our Response: We have revised some sentences to reflect policy implications. At first, we captured only land management implications and we thought that was enough. Our discussions and conclusions are now highlighting “policies”.
- The paper made some errors, including spelling mistake, e.g., “infleunce”(see line 74), “definitiions” (line 246), and incomplete sentences, e.g., lines 106-107.
Our Response: We understand there are some errors like these, and we have corrected all of them.
Reviewer 3 Report
The authors has addressed most of my comments raised for improvement of the first version of the manuscript. However, I'm not very happy with the answer of the authors concerning several comments " these were beyond the scope of the study, we did not consider them".
Author Response
Reviewer 3 Comments and Suggestions for Authors
The authors has addressed most of my comments raised for improvement of the first version of the manuscript. However, I'm not very happy with the answer of the authors concerning several comments " these were beyond the scope of the study, we did not consider them".
Our Response: Thank you. We responded candidly to your comments in the first round review based on the scope research.
Reviewer 4 Report
Most of the concerns have been addressed by the authors. The paper is more clear now.
Author Response
Reviewer 4 Comments and Suggestions for Authors
Most of the concerns have been addressed by the authors. The paper is more clear now.
Our Response: Thank you.